# Nonlinear Chiral-like State Transfer realized with a minimal set of parameters

Kai Bai [1], Chen Lin[1], Tao Liu[1], Jia-Zheng Li [1], Xin Lyu[1] & Meng Xiao [1,2] ✉

Chiral state transfers (CSTs) associated with exceptional points are always under the scrutiny of theoretical and experimental science, owing to its exotic physics and fascinating potential applications. In conventional wisdom, CSTs require adiabatically steering the non-Hermitian Hamiltonian in the parameter space, which inevitably leads to ultra-complex experimental setups and long evolution times, thereby exhibiting a bottleneck in integrating them into compact devices. Here, through both theoretical analysis and circuit-based experiments, we demonstrate nonlinear-exceptional-point-associated chiral-like state transfer, wherein the final state depends on the direction of parameter steering. Notably, our scheme does not require adiabatic evolution. In particular, the steering trajectory can be effectively reduced to three distinct points by varying a single parameter in the parameter space, thereby significantly lowering the experimental complexity. We show that these nonlinear chiral-like state transfers (NCSTs) in our system strictly arise from the basins of attraction for the stable states of nonlinear Hamiltonians, and have no counterparts in linear (non-) Hermitian systems. Our finding transforms the fundamental understanding of CSTs, and may open up new avenues for miniaturizing various relevant applications.

Exceptional points (EPs) are unique spectral singularities in non-Hermitian systems at which two or more eigenvalues and their corresponding eigenvectors coalesce. Recent advances associated with EPs[1–3] have significantly enriched our understanding of non-conservative elements such as gain, loss, and non-Hermitian couplings[4,5]. Slowly encircling an EP in the parameter space inevitably leads to a chiral state transfer (CST) where the final state after an encircling period depends only on the encircling direction, either clockwise or counterclockwise[6]. In the past few years, this fascinating feature has garnered considerable attention from both theoretical and experimental realms[7–24]. For instance, ref. 15 proposed maximizing the difference between decay rates within the shortest possible time in order to enhance the population that remains in the desired state by the end of the encircling process. The flourishing of this field owes not only to its physical elegance (irrelevant of encircling details) and counterintuitive nature (lacking Hermitian counterparts), but also to its great potential for quantum information processing and quantum

communication[25], topological energy transfer[7], polarization state conversion[8], and on-chip optical devices such as optical isolators[21,22] and directional lasing[23], etc[26,27].

The core mechanism of the CSTs demonstrated in previous works lies in the nonadiabatic transitions (NATs) encountered in the adiabatic parameter steering process. Recently, the parameter steering paths have also been extended to open paths[14,19,24] and paths excluding EPs (still in proximate to the EPs)[18,28]. The conceptual evolution of CST is provided in the Supplementary Note 1. However, adiabaticity, as a prerequisite for parametric loops, is still demanded. In practical schemes, adiabaticity indicates a long enough evolution time or distance in the coupled waveguide systems[17,18], which unfortunately, typically results in low transmission efficiency due to path-dependent losses and difficulty in preserving the wave amplitude and noise within acceptable ranges[11,12,17,21,29]. Moreover, a controllable continuous path (including open one) in the parameter space requires a sophisticate experimental setup, which further hinders the miniaturization of the

[1]Key Laboratory of Artificial Micro- and Nano-structures of Ministry of Education and School of Physics and Technology, Wuhan University, Wuhan 430072, China. [2]Wuhan Institute of Quantum Technology, Wuhan 430206, China. ✉e-mail: phmxiao@whu.edu.cn

devices exhibiting CSTs and limits their potential usage in highly integrated platforms. Clearly, breaking adiabaticity and realizing CSTs in a simpler and more efficient manner will be of great interest in pursuing potential applications related to CSTs.

Recently, considerable efforts have been invested to see the impacts of nonlinearity on non-Hermitian systems. In nonlinear non-Hermitian systems, CSTs between bistable states[30,31] can also occur when adiabatically encircling a unique spectral singularity, a nonlinear EP (NEP)[31–33]. Thanks to the feedback mechanism of nonlinear saturable gain and the preservation of a complete basis in dynamics, the inherent noise, which diverges at linear EPs, is greatly suppressed in the parameter steering process around a NEP[34–37]. More intriguingly, benefiting from the feedback mechanism, the instantaneous mode amplitudes are preserved within a small range[31], thus ensuring a mode-independent high CST efficiency. These merits of CSTs at NEPs are highly desirable for the miniaturization of the relevant key applications. However, the physical mechanism underlying NEP-associated CSTs remains unclear. Going one step further, one may also wonder: can NEP-associated CSTs surmount the prerequisite of adiabaticity?

Here, we theoretically elucidate and experimentally validate the unique physical mechanism of the CSTs at a third-order NEP (NEP$_3$) based on two coupled resonators combined with a nonlinear saturable gain. At the NEP$_3$, two stable modes and one unstable mode coalesce. Due to the feedback mechanism of the saturable gain, any random initial state will be attracted to one of the stable modes in a short time. Thus, these two stable modes function both as self-consistent eigenmodes of the nonlinear Hamiltonian and as attractors in the parameter space. We show that, different from NAT-based CSTs in linear systems, the CSTs demonstrated herein strictly arise from the basins of attraction for stable states, which is hence not limited by adiabaticity. Thus, we introduce the term "nonlinear chiral-like state transfer (NCST)" to highlight the underlying mechanism. Intriguingly, the trajectory of the parameter steering process in our system does not necessarily need to include the NEP$_3$. More interesting, it can even be simplified to three distinct points by varying a single parameter in the parameter space. This excellent merit lacks linear (non-) Hermitian counterparts and can significantly reduce the complexity of experimental setups. We have examined this new physical mechanism with an electronic circuit comprising two LC resonators with an electronically controlled three-port switch for selecting the targeting inductors. Our experiments demonstrate unique features of CSTs in nonlinear systems, and point to significant potential in miniaturizing various relevant key applications.

## Discussion

Our system is sketched in Fig. 1a. The dynamics of the system, as the $2 \times 2$ Hamiltonian $H(t)$ changes in time is described by the time-domain Schrödinger equation

$$i\partial_t \left| \boldsymbol{\psi^R(t)} \right\rangle = H(t) \left| \boldsymbol{\psi^R(t)} \right\rangle \quad (1)$$

where $i$ is the imaginary unit, $\left| \boldsymbol{\psi^R(t)} \right\rangle = (\psi_A, \psi_B)^T$ is the state at time $t$ with the superscript T short for transpose, and $\psi_A \equiv |\psi_A|e^{i\theta_A}$ and $\psi_B \equiv |\psi_B|e^{i\theta_B}$ represent the field of the red (left) and blue (right) resonators A and B, respectively. The nonlinear Hamiltonian $H_{|\psi^R\rangle}$ is

$$H(t) = \begin{pmatrix} \omega_A + ig_A(|\psi_A|) & \kappa \\ \kappa & \omega_B - il_B \end{pmatrix}, \quad (2)$$

where $\omega_A$ and $\omega_B$ are the corresponding resonant frequencies, respectively. $g_A(|\psi_A|)$ denotes a nonlinear saturable gain which decreases with the increasing of $|\psi_A|$. $l_B$ represents the linear loss in resonator B, and $\kappa$ is the coupling strength. For simplicity, we set $\kappa$ to 1, and set $\omega_A$ to 0 since any global frequency shift is irrelevant. Letting $g_A$

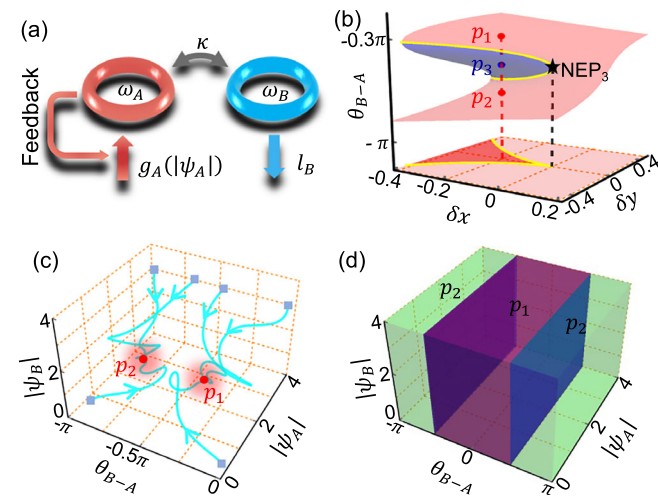

**Fig. 1 | Stability analysis and basins of attraction. a** Schematic of our model. It consists of two coupled resonators with resonance frequencies $\omega_A$ and $\omega_B$, coupling rate $\kappa$, linear loss $l_B$, and nonlinear gain $g_A(|\psi_A|)$ that depends on the field amplitude $|\psi_A|$. **b** Phase differences $\theta_{B-A}$ of the steady modes in the parameter space. **c** Given different initial states (gray squares), the evolution trajectories of system in the phase space. **d** The basins of attraction for the stable modes associated with the $p_1$ (purple domain) and $p_2$ (green domain). The parameters used are: $\omega_A = \omega_B = 0$, $\kappa = l_B = 1$; in (**c**, **d**) $\delta x = -0.2$, $\delta y = 0$ and $g_A = 3/(1 + |\psi_A|^2) - 0.05$.

be a free parameter, the steady modes of the system can be attained, and a NEP$_3$ is achieved at $\omega_B = 0$ and $l_B = 1$[30–34].

Since frequency is not well-defined when the parameter steering is not adiabatic, we choose the phase difference $\theta_{B-A} \equiv \theta_B - \theta_A$ of the state under consideration as an alternative observable. Fig. 1b plots $\theta_{B-A}$ in the parameter space, where $\delta x$ represents the perturbation along the $l_B$ direction (i.e., $1 + \delta x$), and $\delta y$ represents the perturbation along the $\omega_A$ direction (i.e., $0 + \delta y$). The light red and light blue regions represent stable and unstable modes, respectively. These stable and unstable regions are separated by two nonlinear exceptional arcs consisting of NEP$_2$s (yellow lines), which two further merges at the NEP$_3$ (the black star at $\delta x = \delta y = 0$). The plane below shows the projection of the eigenspectrum onto the $\delta x$-$\delta y$ parameter space, where the light red and red regions represent the parameters supporting monostable and bistable modes, respectively. Similar to conventional linear EPs, slowly encircling this NEP$_3$ in the $\delta x$-$\delta y$ parameter space can lead to a NCST, and the final state is also solely determined by the helicity of the parametric loop[30] (see also the Supplementary Note 2.). However, the underline mechanism of the NCSTs in nonlinear non-Hermitian systems remains unclear.

For the nonlinear Hamiltonian considered here, the stable modes not only serve as self-consistent eigenmodes of the Hamiltonian, but also act as attractors in the phase space ($\theta_{B-A}$, $|\psi_A|$, $|\psi_B|$). Specifically, any random initial states will be quickly attracted to one of the stable modes. For a chosen set of parameters in the bistable region [$\delta x = -0.2$ and $\delta y = 0$, as denoted by the red vertical dashed line in Fig. 1b], Fig. 1c illustrates the corresponding trajectories of evolution in the phase space for a few typical initial states (the gray squares represent their positions). The red bullets at $p_1$ and $p_2$ denote the two possible final stable modes. Clearly, in the bistable region, the final state depends strongly on the initial states. The phase space volume that contains the initial states which collapse onto a specific stable mode constitute a basin of attraction, and its size provides a measure of how attractive this stable mode is. Fig. 1d shows the basins of attraction for the two stable modes. If the initial states lie within the purple region of the phase space, the system will evolve towards $p_1$; otherwise, it falls onto $p_2$. This feature, which is unique for nonlinear systems, indicates that achieving a NCST between $p_1$ and $p_2$ can be

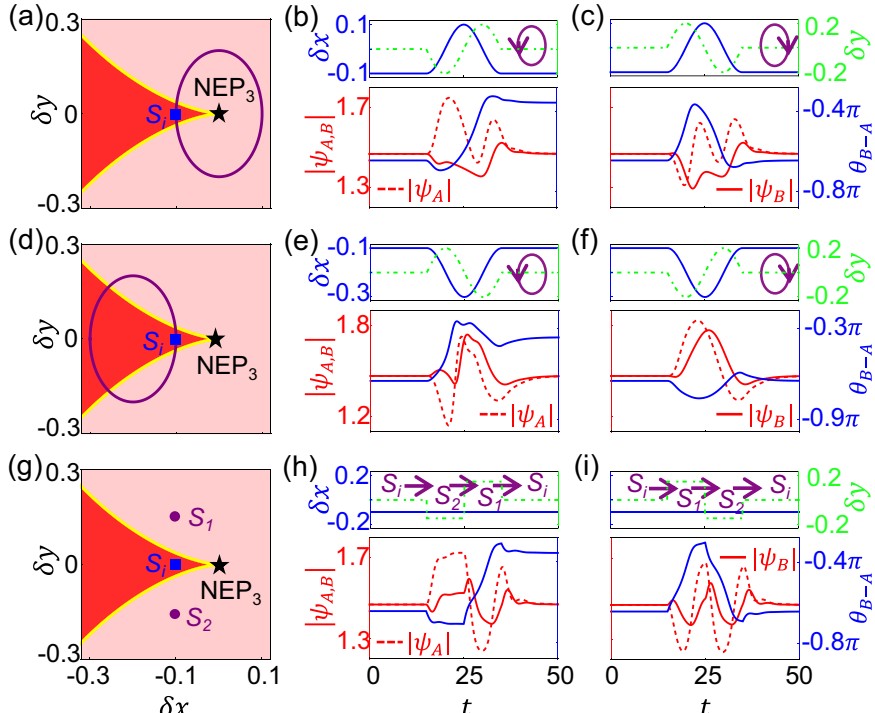

**Fig. 2 | NCSTs with a minimal set of parameters. a–c** A loop centered at the NEP₃. The evolution trajectory of steering parameters in the parameter space **a** and of the states in the phase space (**b, c**). **d-f,** The same as (**a–c**), but now the NEP₃ is not included by the parametric loop. The NCST is retained. **g-i,** To pursue a minimal scheme, the trajectory of the steering process is simplified to three distinct points {$S_i$, $S_1$, $S_2$} by varying a single parameter in the parameter space. $\delta x = -0.1$ and $\delta y = -0.15$, 0, 0.15 for $S_2$, $S_i$ and $S_1$, respectively. The purple circling loops in (**a, d**) are defined in the upper panels of (**b, c**) and (**e, f**), respectively. Other parameters used are the same as those in Fig. 1.

accomplished by selecting the trajectory and speed of the parameter steering process. It is crucial that, as the system evolves along different directions towards the end, the instantaneous states should correspondingly fall into different basins of attraction. Hence, the NCST discussed here strictly arises from the contribution of the attractor properties of the bistable modes. Therefore, achieving NCSTs in nonlinear systems can be free of the requirements in linear systems such as the encirclement of an EP, adiabatic parameter steering, a continuous path or experiencing NATs.

To unveil this fact more explicitly, we demonstrate the NCST associated with this NEP₃ in Fig. 2. The system starts with the initial state in the bistable region (marked by the blue squares) and at a lower phase difference stable state ($\theta_{B-A} = -0.64\pi$). For a loop centered at the NEP₃ [see Fig. 2a], Fig. 2b, c shows the evolution trajectory of steering parameters {$\delta x$, $\delta y$} in the parameter space (upper panel), and the field amplitudes {$|\psi_A|$, $|\psi_B|$} and relative phase $\theta_{B-A}$ (lower panel) as functions of $t$. Here, adiabaticity is not a prerequisite for encirclement, thus the steering process no longer needs to be carried out slowly. In fact, the steering speed can be substantially increased, and the system —except at $S_i$—does not need to evolve toward its corresponding stable state. For instance, the time consumed in Fig. 2 is orders of magnitude smaller than that in Supplementary Fig. 2. The state evolves towards the higher phase difference state ($\theta_{B-A} = -0.36\pi$) in the anticlockwise direction [Fig. 2b] and the initial state in the opposite direction [Fig. 2c], thus exhibiting a NCST between the two stable states. Intriguingly, the evolution towards the initial state exhibit no obvious NATs [see also Supplementary Fig. 2b], which is different from previous demonstrations in both linear[7,8,17–23,28] and nonlinear[30,31] systems. Fig. 2d–f shows that even by excluding the NEP₃ from this single-cycle parametric loop, the chirality of the process can still be retained without NATs. It is worth noting that, compared to the adiabatic case, the chirality here exhibits a "reversal" (see Supplementary Note 3).

Enlighted by Fig. 2d–f, we proceed to further simplify the parameter steering process as a single-cycle parametric loop still requires tedious and complex design. Clearly, it would be of great interest to achieve NCSTs within a minimal set of parameters. Fig. 2g–i shows that NCSTs can indeed be achieved using even only three points {$S_i$, $S_1$, $S_2$} in the parameter space considering the order of $S_1$ and $S_2$. This highlights that NCST can be achieved by varying only a single component, which can significantly simplify the experimental setup (see Supplementary Note 4 for details) while maintain reliable predictability (see Supplementary Note 5 for details). In addition, rather than exponentially growing or decaying in linear systems, the instantaneous wave amplitudes in all demonstrations in Fig. 2 are preserved within a small range. (See also the simulations for the higher phase difference initial state in Supplementary Note 6.)

Fig. 3a shows a circuit consisting of two LC resonators coupled by a capacitor $C_c$ to implement the minimal scheme presented in Fig. 2g–i. The LC resonator on the right hand side is lossy with a normal resistor $R_B$, while the other one possesses an effective negative resistor $-R_A$ that exhibits a saturable gain[31–33]. As outlined by the left dashed block, the effective negative resistor $-R_A$ consists of a series of resistors {$R_1$, $R_2$, $R_g$}, a voltage amplifier $A$ and a two-diodes $D$. The resistance $R_D$ of the two-diodes is a monotonic decreasing function of the voltage applied on it[33]. For an ideal amplifier, $-R_A = -(R_2 R_1 R_g/R_D)/(R_2 \cdot R_1 R_g/R_D)$ and as a result, the negative resistance is a monotonic decreasing function of $|V_A|$ (see the Supplementary Note 7). Here $R_2$ plays the role of intrinsic loss rate and the nonlinear saturation arises from the diodes. Assume that the circuit is working with a time-harmonic field $e^{-i\omega t}$, and the Kirchoff's equations of the circuit are

$$\frac{V_A}{-i\omega L_A} - \frac{V_A}{R_A} - i\omega C_0 V_A - i\omega C_c(V_A - V_B) = 0,$$
$$\frac{V_B}{-i\omega L_B} + \frac{V_B}{R_B} - i\omega C_0 V_B - i\omega C_c(V_B - V_A) = 0. \tag{3}$$

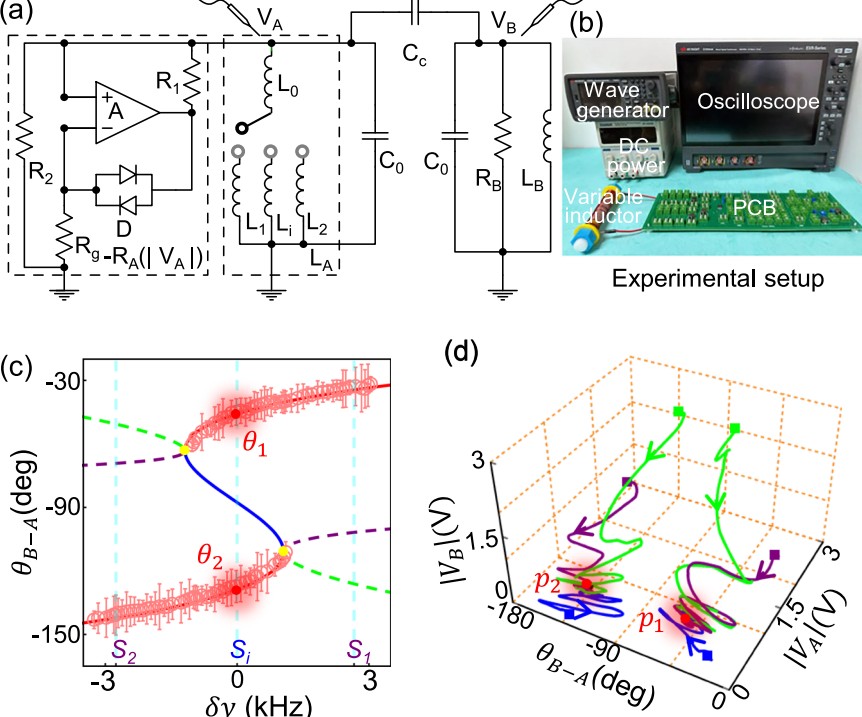

**Fig. 3 | Experimental observation of evolution trajectories in the phase space.** **a** The circuit used in the experiment, showing the inductors (L), capacitors (C), resistors (R), diodes (D), and an amplifier (A). The left black dashed rectangular marks the negative resistor $-R_A$ ($|V_A|$), while the right one marks the inductive element $L_A$, the value of which is selected by the triple-throw switch. **b** A photo of the experimental setup. **c** The relative phase difference $\theta_{B-A} = \arg(V_B) - \arg(V_A)$ versus the perturbation $\delta y$ by varying $L_A$. Here, the perturbation $\delta y$ is defined as $\delta y \equiv (\omega_A - \omega_{A0})/2\pi$, where $\omega_{A0}$ represents the resonance frequency of resonator $A$ when $L_0$ is connected in series with $L_i$ (the effective inductance marks $L_{AO}$). The measured values are averaged over 8 independent measurements. The experimental errors (standard deviation over these 8 measurements) are smaller than the marker size. For demonstration purposes, we exaggerate the error bars by a factor of five. **d** The measured evolution trajectories in the phase space with different initial states (marked by the colored squares). In (c), $S_i = 0$, $S_1 = 2.66$ and $S_2 = -2.76$. In (d), $\delta y = S_i = 0$. Parameters used in the experiments are $L_{AO} = 229.1\ \mu H$, $C_O = 18.3$ nF, $Cc = 4$ nF, $R_B = 945.1\ \Omega$, and $L_B = 226.9\ \mu H$. The detail property of $R_A$ is calibrated in S.M. Sec. 4 of ref. 32.

Voltages $V_A$ and $V_B$ represent the fields inside the left and right resonators, respectively. If we assume $C_c \ll C_0$ and $|\omega_{A,B} - \omega| \ll \omega$, where $\omega_{A,B} = 1/\sqrt{L_{A,B} C_0}$ represents the resonant frequency of the uncoupled resonator, then Eq. (3) becomes

$$\begin{pmatrix} \omega_A + i/2C_0 R_A & \omega_B C_c/2C_0 \\ \omega_B C_c/2C_0 & \omega_B - i/2C_0 R_B \end{pmatrix} \begin{pmatrix} V_A \\ V_B \end{pmatrix} = \omega \begin{pmatrix} V_A \\ V_B \end{pmatrix}, \quad (4)$$

Compared with Eq. (2), the coupling, loss, and saturated gain are given by $\kappa = \omega_B C_c/2C_0$, $l_B = 1/2C_0 R_B$, and $g_A = 1/2C_0 R_A$, respectively. The three points in the parameter space are selected by tuning the inductor $L_A$, which consists of two inductors connected in series: $L_0$ and one of either $L_1$, $L_2$ or $L_i$. Fig. 3b shows our experimental setup. The oscilloscope records the waveforms of $V_A$ and $V_B$, which then gives us the corresponding frequencies, voltages, and relative phases. In the experiment, the specific inductance of $L_A$ is selected by a triple-throw switch which is implemented using two single-pole-double-throw switches and controlled by an arbitrary waveform generator (see "Methods" and Supplementary Note 8). The arbitrary waveform generator also generates the required external driving signal to excite circuits. The dc power supplies power for the amplifier and the two switches. We add a homemade variable inductor (lower left) to fine-tune $L_A$ to control the resonance frequency of resonator A. Details of the circuit elements on the printed circuit board (PCB) can be found in Supplementary Note 9.

We start with the calibration of the stable modes. We record the corresponding relative phase [see Fig. 3c], the frequency [see Supplementary Fig. 12d] and the ratio of voltage [see Supplementary

Fig. 12f]. The open circles are measured results, and the red and blue solid lines represent the stable and unstable steady states obtained with Eq. (3), respectively. As there are active components (amplifier and switches) in the circuit, after we tune on the dc power supply to provide the corresponding operating voltages, the system will automatically reach and stay at a stable mode. To demonstrate the attractor properties of the bistable modes, we applied an external single-frequency driving signal on $V_A$, waited till the circuit's working frequency was the same as the driving signal, and marked the state as the initial state [see the colored squares in Fig. 3(d)]. After that, we removed the external driving signal and started to record the evolution of the system (details provided in Supplementary Note 11). Fig. 3(d) shows the corresponding evolution trajectories in the phase space for a few typical initial states. Clearly, in the bistable region, the final state depends strongly on the initial conditions, which is crucial to achieve NCSTs.

We then proceed to show the NCST between bistable states. We use the arbitrary waveform generator to produce a voltage that controls the value of $\delta y$. To be more specific, the generated voltage controls the switch such that $L_0$ is connected in series with $L_i$, $L_1$ and $L_2$ separately, corresponding to $\delta y$ being equal to $S_i$, $S_1$, and $S_2$, respectively. (Details provided in Supplementary Note 12). Fig. 4a, b show the evolutions of $\delta y$ for different direction of the steering process. At the same time, we record the waveforms as shown in Supplementary Fig. 17. From those, we obtain the evolution of the amplitude $|V_{A,B}|$ (red lines) and the relative phase $\theta_{B-A}$ (blue lines). For the direction of the steering process in Fig. 4a, the state evolves faithfully towards the higher relative phase state independent of the initial state [see

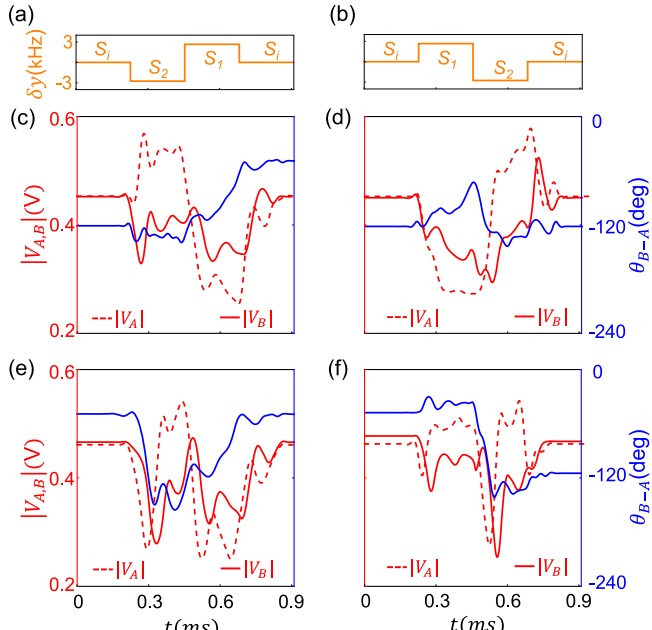

**Fig. 4 | Experimental observation of NCSTs with a minimal set of parameters. a, b** The evolution of $\delta y$, which correspond to the different directions of the steering process. **c, d** Given the initial state as the lower-relative-phase stable state, the state evolves to the higher-relative-phase state for $\delta y$ in **a** and to itself for $\delta y$ in (**b**). **e, f** Similar as (**c, d**) but taking the initial state as the higher-relative-phase stable state. The definition of $\delta y$ and the parameters used are the same as those in Fig. 3.

Fig. 4c, e]. In contrast, for the opposite direction shown in Fig. 4b, the state ends in the lower relative phase state [see Fig. 4d, f]. These evolutions elucidate the fact that, regardless of the initial state, the system always ends in a state enforced by the direction of the steering process. In other words, we have thus demonstrated the NCST between the bistable states within only three points in the parameter space.

In summary, we demonstrate through both theory and circuit experiments a new mechanism of NCSTs in nonlinear non-Hermitian systems. In contrast to CSTs in linear systems that rely on NATs at adiabatic parameter steering process, here the NCSTs in nonlinear systems strictly arise from the basins of attraction for stable states. In our experiments, we observe the NCST between the bistable states using only three distinct points in the parameter space. Consequently, the experimental setups are significantly simplified and the time consumed during the NCSTs is greatly reduced. Here the NCSTs exhibits strong robustness against perturbations and are insensitive to input states, suggesting promising applications in quantum information, computing, and communications[38,39]. Shedding light on the new possibilities of CSTs in nonlinear systems, our work advances the intersection of nonlinear non-Hermitian physics and provides new strategies for the development of practical devices exhibiting chiral-transmission.

## Methods
### Circuit design and fabrication
The implemented circuit consists of two LC resonators coupled by a capacitor Cc, as shown in Fig. 3a. The LC resonator on the left-hand side has an effective negative resistor $-R_A$. The negative resistance is a monotonically decreasing function of $|V_A|$ and exhibits saturable gain[31-33]. To observe the CSTs using the minimal scheme presented in Fig. 2g–i, the inductive element $L_A$ consists of two inductors connected in series: $L_0$ and one of either $L_1, L_2$ or $L_i$. The specific inductance of $L_A$ is selected by a triple-throw switch, implemented using two single-pole-double-throw switches and controlled by an arbitrary waveform generator. Here, $L_0, L_1$ and $L_i$ are standard inductors, Murata

MDH12577C-221MA, MDH12577C-330MA and MDH12577C-150MA, with measured inductances of 210.1uH, 32.2uH and 15uH, respectively (at 70 kHz and 0.5 V). $L_2$ is a wire with a measured inductances of 0 uH. The single-pole-double-throw switch is the ADG1519. Fig. S4(a) presents the functional block diagram of the ADG1519. When VIN is below 0.8 V, terminals D and SA are connected; when VIN exceeds 2 V, conduction occurs between terminals D and SB. Thus, we can use the arbitrary waveform generator to produce a voltage that controls the switch such that $L_0$ is connected in series with $L_i, L_1$ and $L_2$ separately, corresponding to $\delta y$ being equal to Si, S1, and S2, respectively. Moreover, to fine tune $L_A$, we add a homemade variable inductor [lower left in Fig. 3b of the main text], which is wound with 39 turns of #15 copper wire on a PVC cylinder with diameter 2 cm and length 19 cm. By adjusting the length of the solenoid, the inductance can cover the range of interest.

### Measurement set-up
The values of inductances, capacitances, and resistances were also measured experimentally using a precision LCR meter (TH2829C). The arbitrary waveform generator used is the Keysight 33600 A, the DC power supply is a Zhaoxin ZF-3005D, and the oscilloscope is the Keysight EXR054A.

## Data availability
The data that support the findings of this study are available within the main text and the Supplementary Information file. Additional data are available from the corresponding authors upon request.

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

## Acknowledgements

This work is supported by the National Key Research and Development Program of China [Grant No. 2022YFA1404900 (M.X.)], the National Natural Science Foundation of China [Grants No. 12274332 (M.X.), No. 12334015 (M.X.), 12321161645 (M.X.), 12404440 (K.B.), 12447173 (T.L.)], the China Postdoctoral Science Foundation under Grant No. 2023M742715 (K.B.), the China National Postdoctoral Program for Innovative Talents under Grant No. BX20240266 (K.B.) and Postdoctor Project of Hubei Province under Grant No. 2024HBBHCXB054 (K.B.).

## Author contributions

M.X. and K.B. initiated the project. K.B. and C.L. did the simulations, designed the samples and performed the experiments with help from T.L., X.L. and J.-Z.L. All authors contributed to discussions of the results. M.X. and K.B. wrote the manuscript. M.X. led the research.

## Competing interests

The authors declare no competing interests.
