## [Transparent Peer Review file · Nature Communications]

Nonlinear Chiral-like State Transfer realized with a minimal set of parameters

Corresponding Author: Professor Meng Xiao

Version 0:

Reviewer comments:

Reviewer #2

(Remarks to the Author)
See Attachment

[Editorial Note: This attachment is displayed at the end of the file]

Reviewer #3

(Remarks to the Author)
I co-reviewed this manuscript with one of the reviewers who provided the listed reports. This is part of the Nature Communications initiative to facilitate training in peer review and to provide appropriate recognition for Early Career Researchers who co-review manuscripts.

Reviewer #4

(Remarks to the Author)

This is an interesting work, particularly for experimental reasons, as it enables a chiral state transfer without significant losses of the initial state regardless of the non-Hermitian system under study. The basic theory is well known and was first presented in their Ref. 25. In Ref. 25, it was shown that the time-asymmetric state-exchange mechanism, which is based on our ability to locate non-Hermitian degeneracies, enables one to control the dynamics by external parameters of the fields with which the system under study interacts. The main idea of Ref.25 is to maximize the difference between decay rates in the shortest possible time. They should mention this clearly in their paper. Their main theoretical contribution is demonstrating that the non-linearity in the mechanism enables a large difference between the decay rates of the two coupled states. This in turn allows to achieve efficient chiral state transfer much faster than what can be achieved by applying a linear transfer mechanism as for example short-pulse laser. However, they did not provide an explanation of how the nonlinear, non-Hermitian Hamiltonian given in Eq. 2 was derived. I recommend accepting the paper for publication after modifications, where the two points mentioned above are clarified.

Version 1:

Reviewer comments:

Reviewer #2

(Remarks to the Author)

The authors have made significant efforts to improve the manuscript and have addressed many of my previous concerns. However, I still have a few comments regarding their reply.

In both the linear and nonlinear cases of adiabatic parameter-steered chiral state transfer (CST), the final state after a full encircling process typically depends only on the direction of encircling—either clockwise or counterclockwise. However, in the non-adiabatic case, or in the simplified case with only three points, the behavior appears to be quite different.

First, although the final states differ depending on the direction of encircling, the results appear unpredictable. As the authors noted in the rebuttal letter (see Fig. R3 and Fig. R5), the final states depend not only on the steering speed but also on the specific trajectory taken. This stands in sharp contrast to adiabatic CST, where the final outcome is determined solely by the direction of encircling.

Second, the authors claim in the main text that "...the steering speed can be substantially increased, and the system does not need to evolve toward its corresponding stable state." This suggests that the final state after the encircling process may not correspond to any of the system's stable states. In such a case, the process cannot be interpreted as a chiral state transfer between two bistable states.

Based on these two points, the use of the term "CST" in the non-adiabatic case is questionable, as it appears to sacrifice the essential chiral and deterministic nature of the adiabatic CST, and instead merely exhibits differences in the final state without a clear underlying mechanism or reliable predictability. Indeed, I suspect that the underlying mechanism of the observed "CST" in the non-adiabatic case is but the remnant of the adiabatic case, both related to the bistable spectrum. Another relevant comment is that, similar non-adiabatic process should also emerge in the Liouvillian case, though may not be easily observable in Rydberg gases due to the fast relaxation therein.

Finally, I believe that some of the additional results presented in the rebuttal letter—such as Fig. R1 and Fig. R3—are important for understanding the authors' claims. These figures should be included in the Supplementary Note to provide a more complete picture of the system's behavior.

Reviewer #3

(Remarks to the Author)

Reviewer #4

(Remarks to the Author)

The detailed derivation of the non hermitian non linear equations is what is blocked for. Recommend publication.

Version 2:

Reviewer comments:

Reviewer #2

(Remarks to the Author)

The authors have updated the terminology to "nonlinear chiral-like state transfer" and included the corresponding explanations in the main text. As all my questions are clarified, I would like to recommend the work for publication in Nature Communications.

Minor question:

Should the labels in Fig. S6 (l) and (o) be $S_1 S_2$ rather than $S_2 S_1$?

Reviewer #3

(Remarks to the Author)

Response to the Reviewer Comments

We thank the reviewers for reviewing our manuscript, their comments, and constructive suggestions.

1 Response to the Frist Reviewer -- NCOMMS-24-55305-T

In the manuscript entitled "Chiral State Transfer Realized with a Minimal Set of Parameters," the authors study the physical mechanism of chiral state transfer (CST) near a third-order NEP (NEP3) in a nonlinear non-Hermitian system. They demonstrate CST without the prerequisite of adiabaticity through both theory and circuit experiments. They show that the CSTs in the system strictly arise from the basins of attraction for the stable states of nonlinear Hamiltonians. Furthermore, they simplify the trajectory of the steering process to three distinct points on a straight line in the parameter space and claim that they achieve CSTs with a minimal set of parameters. The work is well-organized and well-written, and it has the potential to generate significant interest in the field of non-Hermitian physics. I have the following concerns and comments, before I can make a recommendation.

Reply 1: We sincerely thank you for your great effort in reviewing our manuscript and a nice summary of our work. Below, we provide detailed responses to the concerns and comments, which are very insightful and constructive in improving our manuscript.

1. The authors claim that they achieve CSTs with a minimal set of parameters, as the results shown in Fig. 2 (g-h) and Fig. 4. They simplify the trajectory of the steering process to three distinct points $\{S_i, S_1, S_2\}$ on a straight line in the parameter space. However, when referring to chiral state transfer (CST), it means that the final state after an encircling period depends only on the direction of encirclement, either clockwise or counterclockwise. Since the three points lie on a straight line, making the trajectory

neither clockwise nor counterclockwise, it's difficult to claim the process chiral, rather than directional.

Reply 2: We thank you for bringing up this critical point. Below, we elaborate on our rationale for describing the phenomena observed in Fig. 2(g–h) and Fig. 4 as chiral state transfers (CSTs).

The quantum adiabatic theorem, a seminal result in quantum mechanics, states that for an infinitely slow parametric perturbation there is no possibility of a quantum jump. With the rapidly growing interest in non-Hermitian systems, Ref. [1] first proposed that when system parameters are adiabatically varied along a loop enclosing an exceptional point (EP), nonadiabatic transitions (NATs) inevitably occur. As a result, for any initial state that is a linear combination of the two eigenstates, the final state after one encircling cycle depends solely on the direction of encirclement—clockwise or counterclockwise. This dependence of the final state on the encircling direction is referred to as CST, as you have pointed out. Since then, this phenomenon has attracted considerable attention from both theoretical and experimental communities—not only due to its physical elegance and counterintuitive nature, but also because of its great potential in quantum information processing, quantum communication, and on-chip photonic devices.

In recent years, with the introduction of new concepts, the definition of CST in non-Hermitian systems has continued to evolve. For instance, the parameter trajectories that give rise to this peculiar dependence of the final state have been extended to include those that do not encircle an EP^{2,3}, as well as open paths that do not form a closed loop^{4–6}. In particular, the path proposed in Ref. [6] is reduced to a straight line with an additional single point in the parameter space. It is worth noting that, for an open path, *the notion of clockwise or counterclockwise evolution is no longer well-defined in the conventional sense*. Nevertheless, this phenomenon continues to be referred to as CST in the literature. Taking this a step further, the concept of CST can be generalized from open paths to a set of discrete points (e.g., three points) in parameter space. In essence, the core mechanism of CST is the controllable NATs during the parameter sweep.

Nonlinear EPs (NEPs) are unique spectral singularities in nonlinear non-

Hermitian systems. During the adiabatic encirclement of NEPs, the final states depend exclusively on the direction of encircling [see Supplementary Fig. 2]. This phenomenon is also commonly referred to as CST in the literature⁷. Here in this work, we show that, different from the NAT-based CSTs in linear systems, this CST phenomenon originates strictly from the basins of attraction for stable states. For clarity, we refer to this phenomenon as **nonlinear CST (NCST)**, with the term “nonlinear” emphasizing its distinct underlying mechanism. For NCST, the parameter trajectories that give rise to the direction-dependent final state can also be extended to include paths that do not encircle a NEP [see Fig. 2(d-f)], as well as open paths and even discrete parameter points.

Focusing on the open paths, the trajectory of the steering process can be simplified to three distinct parameter points $\{S_i, S_1, S_2\}$ (see Fig. R1). The initial point S_i is required to lie within the bistable region (see the blue squares in Figs. R1 with $\delta_x = -0.1$), ensuring that the system evolves in different directions, the instantaneous state can fall into different basins of attraction. *The intermediate points $\{S_1, S_2\}$, however, are not subject to strict constraints.* For demonstration purposes, we systematically vary the positions of $\{S_1, S_2\}$: initially placing them to the right of the NEP₃ [i.e., $\delta_x > 0$, see Figs. R1(a)], moving them to the region between S_i and the NEP₃ [i.e., $-0.1 < \delta_x < 0$, see Figs. R1(d)], then aligning them with S_i along a vertical line [i.e., $\delta_x = -0.1$, see Figs. R1(g)], and finally placing them to the left of S_i [i.e., $\delta_x < -0.1$, see Figs. R1(j)]. Figures R1(b, c), R1(e, f), R1(h, i), and R1(k, l) show the corresponding parameter and state evolution for each case. Clearly, this dependence of the final state is faithfully preserved during the parameter variation.

Given the conceptual evolution outlined above, it is natural to extend the definition of CST to encompass the three discrete parameter points illustrated in Fig. R1(a, d, j). The case where the three points lie on the same line [Fig. R1(g)]—which, as will be discussed below, significantly simplifies the experimental setup—can be regarded as a special case of three discrete points. In the revised manuscript, we use the term *nonlinear chiral state transfer (NCST)* with the word “nonlinear” to acknowledge the different underlying mechanism.

Fig. R1 | NCSTs under different configurations. To achieve NCST, the initial point S_i must lie within the bistable region; however, the points $\{S_1, S_2\}$ are not subject to strict constraints. The first column illustrates different configurations of $\{S_1, S_2\}$; while the second and third columns show the corresponding evolutions of the parameters in parameter space (upper panels) and the system states in phase space (lower panels), respectively. In (a, d, g, j), we fix $\delta y = 0.15$ and -0.15 for S_1 and S_2 , respectively. The values of δx for $\{S_1, S_2\}$ are set to be 0.05 in (a), -0.04 in (d), -0.1 in (g) and -0.16 in (j). In the tight-binding model used in our work, NCST can be realized by varying only δy . In the experimental implementation, δx and δy are generally not independent. In (m), we assume $\delta x = -0.1 + 0.2 \eta + 0.3 \eta^2$ and $\delta y = \eta$, with $\eta = 0.15$ for S_1 and $\eta = -0.15$ for S_2 . Other parameters used are the same as those in Fig. 2.

In our work, we consider the case where the three points lie on a straight line in the parameter space to illustrate that *NCST can be achieved by varying only a single component*, which further simplifies the experimental implementation. In the tight-binding model used in our work, changing a single component corresponds to a straight line in the parameter space. However, in practical experimental settings, each external tuning knob typically affects multiple parameters of the tight-binding model. For instance, in our circuit implementation, adjusting a component such as the capacitance C_0 typically leads to simultaneous changes in both the resonant frequencies of the cavities and the effective loss rates in the equivalent tight-binding model [see Eq. (4) in the main text]. In coupled optical waveguides^{8,9} and microcavities^{10,11}, the tuning of the resonance frequency can also cause a change in the coupling strength. Thanks to the absence of strict constraints on $\{S_1, S_2\}$, NCST can be realized by varying only a single external tuning knob. To further illustrate this, we randomly choose $\delta x = -0.1 + 0.2 \eta + 0.3 \eta^2$ and $\delta y = \eta$, where η corresponds to the variation of a single physical component. Figure R1(m-o) shows that this dependence of the final state is also faithfully preserved. Note here, the three points $\{S_i, S_1, S_2\}$ are not on the same straight line in the parameter space though we only tune a single knob, i.e., η .

In summary, the concept of CST has evolved over time and has already been extended to open-path protocols, where the requirement for strict clockwise or counterclockwise encircling has gradually diminished in significance. Here in this work, leveraging the properties of nonlinear systems, we further extend the definition from open paths to include **discrete points** in parameter space. Nevertheless, the dependence of the final state on the direction of parameter variation remains the core feature pursued by both theoretical and experimental researchers. In light of this, and considering the intrinsic nonlinearity of the system, we respectfully propose to refer to this direction-dependent behavior of the final state as NCST, which can be realized by varying only a single external tuning knob.

To clarify this critical point, we have changed the title and the relevant section in the main text. In addition, we have also added a new Note in Supplementary Information (Note 1) to illustrate the conceptual evolution of CST. (please see the

highlighted lines 18-21 and 46-47 in the main text, highlighted Supplementary Note 1)

Title: “*Nonlinear Chiral State Transfer realized with a minimal set of parameters*”

Lines 18-21: “..... *nonlinear-exceptional-point-associated CSTs with no adiabaticity prerequisite through both theory and circuit experiments. three distinct points by varying a single parameter in the parameter space*”

Lines 46-47: “*The conceptual evolution of CST is provided in the Supplementary Note 1.*”

2. Since any random initial states will be quickly attracted to the ‘nearest’ stable modes, my question is whether the evolved state is always restricted to the stable state? If so, the final state should depend solely on the last step, i.e., S1 to Si or S2 to Si. In the first case, the final state always ends at the upper stable state, and in the second case, the final state always ends at the lower stable state.

Reply 3: We thank you for bringing up this point. We completely agree with your insightful observation that “any random initial states will be quickly attracted to the ‘nearest’ stable modes”. For this reason, during an **adiabatically** dynamical encirclement of the NEPs, the evolved state is always restricted to the stable state [see Fig. R2 (b, e) and Supplementary Fig. 2]. Along the clockwise direction [see Fig. R2(a)], the state experiences a nonadiabatic transition at t_1 and evolves to the same lower-phase-difference state from where it starts [see Fig. R2(b)]. Along the anticlockwise direction [Fig. R2(d)], the state stays on the steady stable state surface at all times, evolving adiabatically from a lower-phase-difference state to a higher-phase-difference state [see Fig. R2(e)]. Here the encircling period $T = 5000$, and the encirclement can be regarded as adiabatic. These evolutions elucidate the fact that, provided the circling process is slow enough, dynamical encirclement of a NEP₃ in the parameter space can lead to CSTs and that the evolved state is restricted to the stable state except at the points where nonadiabatic transitions occur. Under the assumption of adiabatic parameter steering, you are absolutely right that “the final state depends solely on the last step”.

However, the NCST discussed here arises from the contribution of the attractor

Fig. R2 | NCSTs at different parameter steering speeds. **a-c**, Along the clockwise direction, the evolution trajectory of steering parameters in the parameter space **a** and of the states in the phase space for different steering speeds (**b**, **c**). **d-f**, The same as **a-c**, but for anticlockwise encirclement. Here, the initial state is selected as the lower-phase-difference stable state. In (**b**, **e**), the instantaneous state is restricted to the stable state except at the point (t_1 in **b**) where nonadiabatic transitions occur. In contrast, in (**c**, **f**), the instantaneous state deviates from the stable state for most of the evolution. The other unspecified parameters used are the same as those in Fig. 2.

properties of the bistable modes. When the system evolves in different directions along the trajectory, the instantaneous states correspondingly fall into distinct basins of attraction, thereby enabling the realization of NCSTs. Therefore, NCSTs can be realized without requiring adiabatic parameter steering. *In other words, the steering speed can be substantially increased, such that the system no longer has sufficient time to evolve toward its corresponding stable state.* For instance, we reduce the steering period from $T=5000$ to $T=50$. Figure R2 (**c**, **f**) shows the corresponding evolution trajectories of the states in phase space (see the lower panels), which differ significantly from those in Fig. R2(**b**, **e**). Thus, the instantaneous state deviates from the stable state for most of the evolution. In other words, *the evolved state is no longer restricted to the stable state.*

For some configurations of the encirclement loops, the speed of the parameter steering process greatly influences the outcome, which may result in a “reversal” of

Fig. R3 | “Reversal” of chirality at different steering speeds. The same as Fig. R2, but now the NEP_3 is not enclosed by the parametric loop.

chirality (see Fig. R3) or disappearance of the chiral behavior (see Fig. R4). Figure R3 shows a single-cycle parametric loop that does not enclose the NEP_3 . Along this path, both adiabatic driving [see Fig. R3(d, e) with $T=5000$] and fast steering [see Fig. R3(c, f) with $T=50$] give rise to NCSTs. However, *a closer comparison between Figs. R3(b) and R3(c), as well as R3(e) and R3(f), reveals that the final evolved states differ, despite the same parametric path being followed.* In Fig. R4, the trajectory of the steering process is simplified to three distinct parameter points $\{S_i, S_1, S_2\}$. When the steering

Fig. R4 | Disappearance of the chiral behavior. The same as Fig. R2, but the trajectory of the parameter steering process is simplified to three distinct points $\{S_i, S_1, S_2\}$.

period $T = 50$, NCST is still observed. However, as the steering period is further reduced to $T = 5$, *the chiral behavior is destroyed, despite the same parametric path being followed.*

Moreover, the trajectory of the steering process can also greatly influence the outcome of a parametric loop. The purple solid lines in Fig. R5 (a, d) present a specific trajectory of the steering parameters in parameter space used to connect S_1 and S_2 . Figures R5(b, e) show the corresponding evolution of the steering parameters (the upper panels) and the system states in phase space (the lower panels). For comparison, Fig. R5(c, f) presents the evolution of the system states when only three discrete points are used in the parameter steering process with the same steering period. *A closer comparison between Figs. R5(b) and R5(c), as well as R5(e) and R5(f), reveals that the trajectory of the steering process also plays a crucial role in determining the chiral behavior. In other words, the final state is not solely determined by the last parameter point.*

Fig. R5 | Influence of different trajectories connecting discrete points. a, d, The trajectory of steering parameters in the parameter space with different directions. **b, e,** The corresponding evolution of parameters (upper panels) and states (lower panels). **c, f,** The same as **b, e**, but with the system parameters abruptly switching between S_1 and S_2 , effectively reducing the trajectory to just three discrete points. The unspecified

parameters used are the same as those in Fig. 2.

In summary: The NCST discussed here originates from the attractor properties of the bistable modes. It is not subject to the adiabaticity prerequisite; hence, *the steering speed can be substantially increased (which is one of the targets in our setup), such that the system does not have sufficient time to evolve toward its corresponding stable state—i.e., the evolved state is no longer restricted to the stable branch.* As demonstrated above, both the trajectory and the speed of the steering process can influence the final state. Therefore, *the final state is not determined solely by the last step of the parameter path.*

To clarify this critical point, we have added a few sentences in the main text. (please see the highlighted lines 153-154 in the main text)
Lines 153-154: “*In fact, the steering speed can be substantially increased, and the system does not need to evolve toward its corresponding stable state.*”

3. Following the concern above, a relevant comment is that the observed ‘CST’ along the straight line is nothing but the hysteresis behavior across bistable transitions in nonlinear systems. Indeed, CST based on the Liouvillian exceptional structure (the open-system counterpart of the eigenmode landscape in Fig. 1b here) of Rydberg vapors has recently been reported (arXiv:2402.02779), wherein the CST along the straight line, as discussed here, is a direct result of the well-known bistability transition of the Rydberg gas.

Reply 4: We thank you for bringing this interesting reference to our attention, which we had not noticed before.

In response, we emphasized that the case where the three points lie on a straight line in parameter space was used to demonstrate that *NCST can be realized by varying only a single external tuning knob*, thereby further simplifying experimental implementation [see **Reply 2**].

With due respect, although the final state at the end of the steering process is

indeed a stable one, the NCST observed here cannot be simply interpreted as a result of conventional hysteresis behavior across bistable transitions. There are several significant differences, and a detailed comparison and the origin of the CST observed in this work have been carefully analyzed in **Reply 3**.

In arXiv:2402.02779, the authors explore the non-equilibrium dynamics of dissipative Rydberg gases within the framework of non-Hermitian physics, and experimentally realize **adiabatic** chiral state switching associated with a Liouvillian EP between two collective steady states. Therein, due to the constraint of the adiabatic steering process, the evolved state is always restricted to the stable state. Based on this, the authors concluded that the observed CST originates from the similarity between the collective steady-state landscape near the Liouvillian exceptional structure and the eigen-spectrum near an EP of a non-Hermitian Hamiltonian.

In our work, NCSTs can be achieved **without the need for adiabatic parameter steering**. Consequently, the parameter steering speed can be substantially increased, such that the system does not have sufficient time to evolve toward its corresponding stable state (see Fig. R3). The behavior of the **eigen spectra alone** cannot capture the rich **dynamical** phenomena associated with NCSTs that emerge beyond the regime of adiabatic parameter steering.

To clarify this critical point, we have incorporated this reference (cited as Ref. [16] in the revised manuscript) and updated the corresponding part of the main text. (please see the highlighted lines 64-66 in the main text)

Lines 64-66: We have revised the original sentence from

“However, there has been NO experimental verification. Going one step further, one may also wonder: can NEP-associated CSTs surmount the prerequisite of adiabaticity?”

to

“However, the physical mechanism underlying NEP-associated CSTs remains unclear. Going one step further, one may also wonder: can NEP-associated CSTs surmount the prerequisite of adiabaticity?”

4. The expression ' $g_A = 3/(1 + |\psi|) - 0.05$ ' in the caption of Fig. 1, should probably be ' $g_A = 3/(1 + |\psi|^2) - 0.05$ '.

Reply 5: We thank you for pointing out this typo, which we have corrected in the revised manuscript.

2 Response to the Second Reviewer -- NCOMMS-24-55305-T

Reply 6: We acknowledge the co-review process and appreciate the thoughtful feedback provided through this collaborative effort. We thank all reviewers, including the Early Career Researcher, for your time and constructive comments that helped improve our manuscript.

3 Response to the Third Reviewer -- NCOMMS-24-55305-T

This is an interesting work, particularly for experimental reasons, as it enables a chiral state transfer without significant losses of the initial state regardless of the non-Hermitian system under study.

Reply 7: We sincerely thank you for carefully reviewing our manuscript. The realization of chiral state transfer without significant loss is one of the key highlights of our work.

The basic theory is well known and was first presented in their Ref. 25. In Ref. 25, it was shown that the time-asymmetric state-exchange mechanism, which is based on our ability to locate non-Hermitian degeneracies, enables one to control the dynamics by external parameters of the fields with which the system under study interacts. The main idea of Ref.25 is to maximize the difference between decay rates in the shortest possible time. They should mention this clearly in their paper.

Reply 8: We sincerely thank you for pointing out the citation error in our manuscript. Based on your description, we assume that you were referring to Ref. [26] in the original manuscript: *Time-asymmetric quantum-state-exchange mechanism*, Phys. Rev. A **88**, 010102 (2013). Indeed, this reference [12] is a foundational work. Historically, Ref. [1] first proposed the concept that when system parameters are adiabatically varied along a loop enclosing an exceptional point (EP), nonadiabatic transitions (NATs) inevitably occur. Subsequently, in Ref. [12], the authors from the same group clarified the underlying mechanism and proposed a protocol to control quantum dynamics in open systems by steering the system around an EP. They emphasized the importance of maximizing the difference in decay rates within the shortest possible time to enhance the population remaining in the desired state at the end of the pulse.

In the revised version, we have corrected this mistake in the citations and added a

few sentences to clarify the contribution of this foundational work. In addition, we have also cited Ref. [1] as Ref. [6] in the revised manuscript. (please see the highlighted lines 35-37 in the main text)

Lines 35-37: “For instance, Ref. [15] proposed maximizing the difference between decay rates within the shortest possible time in order to enhance the population that remains in the desired state by the end of the encircling process.”

Their main theoretical contribution is demonstrating that the non-linearity in the mechanism enables a large difference between the decay rates of the two coupled states. This in turn allows to achieve efficient chiral state transfer much faster than what can be achieved by applying a linear transfer mechanism as for example short-pulse laser. However, they did not provide an explanation of how the nonlinear, non-Hermitian Hamiltonian given in Eq. 2 was derived.

Reply 9: We sincerely thank you for bringing this point to our attention.

After being experimentally demonstrated in microwave cavities¹³, EPs were subsequently observed in various systems, including optical microcavities¹⁴, coupled atom-cavity systems¹⁵, photonic crystal slabs¹⁶, exciton-polariton billiards¹⁷, acoustic systems¹⁸, and others. When two dominant modes are considered in these systems, the coupled mode theory allows one to extract a linear 2×2 non-Hermitian Hamiltonian model, similar to the form given in Eq. (2). In particular, when gain elements are introduced, a realistic gain medium is expected to saturate as the system intensity increases. Therefore, introducing a nonlinear saturable gain model provides a more realistic description. Based on these considerations, we arrive at the nonlinear non-Hermitian Hamiltonian presented in Eq. (2). In optics, a commonly used gain saturation model is $g = \Gamma / (1 + |\psi|^2) - \gamma_0$, where Γ represents the pump strength and γ_0 denotes the intrinsic loss (see Eq.1 in Ref. [19]).

Below, we present a detailed derivation based on the circuit model implemented in our experiment [see Fig. R6(a)]. A similar derivation was also presented in our previous work²⁰. The system consists of two LC resonators coupled with two capacitors C_c . Voltages V_A and V_B represent the fields inside the left and right resonators,

respectively. The nonlinear saturable gain $[-R_A(|V_A|)$, the left dashed block] consists of a voltage amplifier A , a two-diode component D and a series of resistors $\{R_1, R_2, R_g\}$. Here R_2 represents the intrinsic loss rate. Assume that the circuit is working with a time-harmonic field $e^{-i\omega t}$, and Kirchhoff's equations of the circuit are

$$\begin{aligned} \frac{V_A}{-i\omega L_A} - \frac{V_A}{R_A} - i\omega C_0 V_A - i\omega C_c (V_A - V_B) &= 0, \\ \frac{V_B}{-i\omega L_B} + \frac{V_B}{R_B} - i\omega C_0 V_B - i\omega C_c (V_B - V_A) &= 0. \end{aligned} \quad (\text{R1})$$

Assuming $C_c \ll C_0$ and $|\omega_{A,B} - \omega| \ll \omega$, where $\omega_{A,B} = 1/\sqrt{L_{A,B}C_0}$ represent the resonant frequencies of the uncoupled resonators, then Eq. (1) becomes

$$\begin{pmatrix} \omega_A + i/2C_0R_A & \omega_B C_c/2C_0 \\ \omega_B C_c/2C_0 & \omega_B - i/2C_0R_B \end{pmatrix} \begin{pmatrix} V_A \\ V_B \end{pmatrix} = \omega \begin{pmatrix} V_A \\ V_B \end{pmatrix}, \quad (\text{R2})$$

Compared with the tight-binding Hamiltonian in Eq. (2) of the main text, the coupling, loss, and saturated gain terms are given by $\kappa = \omega_B C_c/2C_0$, $l_B = 1/2C_0R_B$, and $g_A = 1/2C_0R_A$, respectively.

Focusing on the effective gain component ²¹[see Fig. R6(b)], the input voltage V_1 at node ① of the operational amplifier is equal to the voltage at node ②. Meanwhile, the current passing through the resistor R_g , $I_{R_g} = V_1/R_g$, must flow out of node ③ and pass through the diodes. Thus, the voltage on node ③, V_3 is given by

$$V_3 = V_1 + \frac{V_1}{R_g} R_D, \quad (\text{R3})$$

Then, the current through the resistor R_1 is given by

$$I_{R_1} = \frac{V_1 - V_3}{R_1} = -\frac{V_1}{R_g R_1} R_D, \quad (\text{R4})$$

Here the “−” sign indicates the current I_{R_1} flows in the direction opposite to the voltage at node ①. Hence from the perspective of the red node in Fig. R6(b), the effective negative resistance (R_n) of the whole system inside the dashed box is $R_n = V_1/I_{R_1} = -R_g R_1/R_D$. Accordingly, we derive the expression

$$-R_A = -(R_2 R_1 R_g/R_D)/(R_2 - R_1 R_g/R_D). \quad (\text{R5})$$

Figure R6(c) shows the volt-ampere curve of a two-diode (Onsemi BAV99L) composite and the corresponding resistance. The resistance of the two-diode R_D is a monotonic decreasing function of the voltage applied on it. Thus, we can obtain that the gain

coefficient $g_A = 1/2C_0R_A$ decreases with the increase of $|V_A|$.

Fig. R6 | Derivation of the nonlinear non-Hermitian hamiltonian. **a**, The circuit used in the experiment, showing the inductors (L), capacitors (C), resistors (R), diodes (D), and an amplifier (A). The left black dashed rectangular marks the negative resistor $-R_A (|V_A|)$. **b**, The nonlinear saturation arises from the two-diode component. **c**, The I-V characteristic curve (blue) and the corresponding resistance (green) of the two-diodes composite (Onsemi BAV99L). The inset shows the measurement setup. **d**, An LC circuit connected in parallel with a normal resistor R and an effective negative resistor (marked by the black dashed box). **e**, Evolution of V_A on the red node in **d** starting with a small kicking-start voltage $V_A = 1 \mu\text{V}$. The solid red lines highlight the envelope. **f**, The effective gain $(2R_A C)^{-1}$ as a function of V_A . The solid line is the fitting curve using Eq. (R7) with fitting parameters: $a = 699$, $b = 18$, and $c = 1$. In (**d**, **e**, **f**), $R_g = 5.1\text{k}\Omega$, $R_2 = 100\text{k}\Omega$, $R_1 = 500\Omega$, $L = 250\mu\text{H}$, and $C = 18\text{nF}$.

We can also obtain the form of the saturation gain from numerical simulations with the circuit shown in Fig. R6(d). We start the simulations with a small initial voltage ($1 \mu V$) on the capacitor to kick start oscillations, and Fig. R6(e) shows the evolution of V_A till a steady state is reached. The corresponding Kirchhoff's equation of the circuit in Fig. R6(d) is

$$\frac{V_A}{-R_A} + \frac{V_A}{-i\omega L} - i\omega C V_A + \frac{V_A}{R} = 0. \quad (R6)$$

The effective resistance $-R_A$ gives the saturable gain of our system. When the voltage reaches a stable value, $R_A = R$ and thus we obtain R_A as a function of the field amplitude (here the voltage at the node A) as shown by the open cyan circles in Fig. R6(f). These data points can be approximated by a fitting curve [the red line in Fig. R6(f)]

$$g(|V_A|) = \frac{1}{2R_A C} = \frac{a}{1 + b|V_A|} - c. \quad (R7)$$

Here, c represents the intrinsic loss rate, a and b are real fitting parameters.

To clarify this critical point, we have included these discussions as a new Note in the Supplementary Information (Note 4).

I recommend accepting the paper for publication after modifications, where the two points mentioned above are clarified.

Reply 10: We sincerely thank you for carefully reviewing our manuscript and recommending this manuscript publication in Nature Communications. The two important points you mentioned above have been addressed one by one in our responses above.

References

1. Uzdin, R., Mailybaev, A. & Moiseyev, N. On the observability and asymmetry of adiabatic state flips generated by exceptional points. *J. Phys. A Math. Theor.* **44**, 435302 (2011).
2. Hassan, A. U. *et al.* Chiral state conversion without encircling an exceptional point. *Phys. Rev. A* **96**, 1–5 (2017).
3. Nasari, H. *et al.* Observation of chiral state transfer without encircling an exceptional point. *Nature* **605**, 256–261 (2022).
4. Li, A. *et al.* Hamiltonian Hopping for Efficient Chiral Mode Switching in Encircling Exceptional Points. *Phys. Rev. Lett.* **125**, 187403 (2020).
5. Shu, X. *et al.* Fast encirclement of an exceptional point for highly efficient and compact chiral mode converters. *Nat. Commun.* **13**, 2123 (2022).
6. Shu, X. *et al.* Chiral transmission by an open evolution trajectory in a non-Hermitian system. *Light Sci. Appl.* **13**, 65 (2024).
7. Wang, H., Assaworrorarit, S. & Fan, S. Dynamics for encircling an exceptional point in a nonlinear non-Hermitian system. *Opt. Lett.* **44**, 638 (2019).
8. Maczewsky, L. J. *et al.* Nonlinearity-induced photonic topological insulator. *Science (80-.)*. **370**, 701–704 (2020).
9. Kirsch, M. S. *et al.* Nonlinear second-order photonic topological insulators. *Nat. Phys.* **17**, 995–1000 (2021).
10. Peng, B. *et al.* Parity–time-symmetric whispering-gallery microcavities. *Nat. Phys.* **10**, 394–398 (2014).
11. Chang, L. *et al.* Parity–time symmetry and variable optical isolation in active–passive-coupled microresonators. *Nat. Photonics* **8**, 524–529 (2014).
12. Gilary, I., Mailybaev, A. A. & Moiseyev, N. Time-asymmetric quantum-state-exchange mechanism. *Phys. Rev. A* **88**, 010102 (2013).

13. Dembowski, C. *et al.* Experimental Observation of the Topological Structure of Exceptional Points. *Phys. Rev. Lett.* **86**, 787–790 (2001).
14. Lee, S.-B. *et al.* Observation of an Exceptional Point in a Chaotic Optical Microcavity. *Phys. Rev. Lett.* **103**, 134101 (2009).
15. Choi, Y. *et al.* Quasieigenstate Coalescence in an Atom-Cavity Quantum Composite. *Phys. Rev. Lett.* **104**, 153601 (2010).
16. Zhen, B. *et al.* Spawning rings of exceptional points out of Dirac cones. *Nature* **525**, 354–358 (2015).
17. Gao, T. *et al.* Observation of non-Hermitian degeneracies in a chaotic exciton-polariton billiard. *Nature* **526**, 554–558 (2015).
18. Tang, W. *et al.* Exceptional nexus with a hybrid topological invariant. *Science (80-.).* **370**, 1077–1080 (2020).
19. Harari, G. *et al.* Topological insulator laser: Theory. *Science (80-.).* **359**, (2018).
20. Bai, K. *et al.* Nonlinearity-enabled higher-order exceptional singularities with ultra-enhanced signal-to-noise ratio. *Natl. Sci. Rev.* **10**, nwac259 (2023).
21. Bai, K. *et al.* Observation of Nonlinear Exceptional Points with a Complete Basis in Dynamics. *Phys. Rev. Lett.* **132**, 073802 (2024).

Response to the Reviewer Comments

We thank the reviewers for reviewing our manuscript, their comments, and constructive suggestions.

1 Response to the Frist Reviewer -- NCOMMS-24-55305A

The authors have made significant efforts to improve the manuscript and have addressed many of my previous concerns. However, I still have a few comments regarding their reply.

Reply 1: We sincerely thank you for your great effort in reviewing our manuscript and are grateful for the recognition of our efforts in improving the manuscript. Below, we carefully respond to the remaining comments point by point.

In both the linear and nonlinear cases of adiabatic parameter-steered chiral state transfer (CST), the final state after a full encircling process typically depends only on the direction of encircling—either clockwise or counterclockwise. However, in the non-adiabatic case, or in the simplified case with only three points, the behavior appears to be quite different.

Reply 2: We fully agree with you that during adiabatic parameter-steered *encircling of an EP or NEP along a closed loop*, the final state typically depends only on the direction of encircling—either clockwise or counterclockwise. We also concur that in the non-adiabatic case, or in the simplified case with only three points, the behavior appears to be different.

We would like to point out that the definition of “CST” has evolved as new features are discovered and new mechanisms are introduced. Notably, the parameter trajectories giving rise to CST have already been extended to include those that do not encircle an EP^{1,2} [see Fig. R1(a)], as well as open paths that do not form a closed loop³⁻⁵[see Fig.

R1(b)]. When the steering process follows a closed trajectory that does not encircle the EP, even under adiabatic evolution, the final state depends not only on the direction of encircling, *but also on the specific geometry of the parametric path* [see Fig. R1(a), as well as Fig. 1 in Ref. [2]]. When the steering process follows an open path [see Fig. R1(b)] proposed in Ref. [5], *the notion of clockwise or counterclockwise evolution is no longer well-defined in the conventional sense*. Nevertheless, in both cases, the term “CST” is still used to describe the associated features.

Fig. R1 | CSTs under different paths. (a), CST without encircling an EP. Chirality map based on the final state after an adiabatic encircling process can be influenced by the detail geometry of the parametric trajectory [see Ref. [2] for details]. (b), CST with an open path. Since the path is not closed, the conventional notion of clockwise and counterclockwise evolution is no longer well-defined [see Ref. [5] for details].

First, although the final states differ depending on the direction of encircling, the results appear unpredictable. As the authors noted in the rebuttal letter (see Fig. R3 and Fig. R5), the final states depend not only on the steering speed but also on the specific trajectory taken. This stands in sharp contrast to adiabatic CST, where the final outcome is determined solely by the direction of encircling.

Reply 3: We thank you for bringing up this critical point. We have carefully considered your comment and have revised the terminology in our work to “nonlinear chiral-like state transfer” to emphasize the distinct underlying mechanism. We acknowledge that the final states depend on both the steering speed and the specific trajectory taken; however, these dependencies are also present in CST reported in previous studies (see Figs. R1 and R2). Below in Table R1, we provide a detailed comparison to clarify the

similarities and differences between EP-based CST and NEP-based nonlinear chiral-like state transfer (NCST), thereby establishing a clear definition of NCST.

Parameter steering property	EP-based CST	NEP-based NCST
Adiabatic; Closed loop including EP/NEP.	Final state is determined solely by the steering direction ⁶ .	Final state is determined solely by the steering direction ⁷ .
Adiabatic; Closed loop excluding EP/NEP.	Final state is determined by both the direction and trajectory ^{1,2} .	Final state is determined by both the direction and trajectory.
Open loop.	Final state is trajectory dependent ³⁻⁵ (see page 3, paragraph 2 of Ref. [4] for a detailed discussion where adiabaticity is assumed).	Final state is determined by both the trajectory and evolution time.
Physical mechanism.	Landscape of eigen-spectrum and non-adiabatic jump .	Landscape of eigen-spectrum and attractor property .

Table R1. Similarities and differences between CST and NCST.

From a theoretical and conceptual standpoint, adiabatic CST schemes based on EP or NEP encirclement are highly appealing due to their simplicity and full predictability (as you have rightly noted), as the final outcome is determined solely by the encircling direction. However, from an experimental perspective, such adiabatic parametric steering processes typically requires long evolution times—which translate to bulky devices in waveguide systems—and extremely complex setups to suppress noise accumulation. This is precisely one of the major issues our NCST aims to address.

There is a general trade-off—relevant to both CST and NCST—between the theoretical preference for adiabatic evolution with full predictability, and the experimental demand for faster evolution to reduce setup complexity and broaden

practical applicability. Hence, reducing the evolution time is desirable, provided that CST can still be stably achieved. Although such schemes inevitably reduce the level of predictability to some extent, as long as the final outcome remains robust within the bounds of experimental uncertainty, the process can still be considered effectively predictable and of practical relevance. This trade-off has already been taken into account in the experimental design of EP-based CST demonstration^{2,8}. Figure R2 presents the dependence of the chirality function—determined by the final state—on the number of roundtrips (i.e., the evolution time) in a fiber-based photonic emulator (adapted from Fig. S2 in Ref. [2]). Thus, in EP-based CST demonstration, the final state also depends on the steering speed and the specific trajectory.

Therefore, from a phenomenological perspective, NEP-based NCST exhibits many similarities with EP-based CST, including a dependence of the final state on both the steering speed and the specific parameter trajectory.

[REDACTED]

Fig. R2 | CSTs under different parametric loops and number of roundtrips (adapted from Fig. S2 in Ref. [2]). Chirality maps based on the final state as a function of the coupling constant and the total number of round trips. (a) The EP (red dot) is located at the center of a circular loop (white circle). (b) The EP lies in close proximity but outside of a circular parametric loop. (c) Parameters vary along an EP-excluding elliptical trajectory.

To clarify this critical point, we have revised the terminology to 'nonlinear chiral-like state transfer' and added corresponding explanatory text in the main manuscript. Additionally, we have adapted Table R1 in the Supplementary Information (Note 1) to highlight the similarities and differences between EP-based CST and NEP-based NCST.

Second, the authors claim in the main text that “...the steering speed can be substantially increased, and the system does not need to evolve toward its corresponding stable state.” This suggests that the final state after the encircling process may not correspond to any of the system’s stable states. In such a case, the process cannot be interpreted as a chiral state transfer between two bistable states.

Reply 4: We thank you for raising this point. Indeed, upon careful reconsideration, we realize that the corresponding sentence in our main text may not have clearly conveyed our intended meaning.

In our nonlinear saturable gain model, when the system parameters are fixed, the feedback mechanism ensures that any random initial state is rapidly attracted to one of the stable modes. These two stable modes thus serve both as self-consistent eigenmodes of the nonlinear Hamiltonian and as attractors in parameter space. This observation motivated us to explore the underlying physical mechanism for realizing CST between two bistable states by engineering the connecting path between initial and final parameter points (which are both S_i), as well as the corresponding evolution time [see Fig. 2 in the main text]. When the system evolves along this path in different directions, the instantaneous states fall into distinct basins of attraction, thereby enabling the realization of NCSTs between these two stable modes at S_i . This new mechanism allows the steering speed in Fig. 2(a) to be significantly increased, since during the parameter evolution, the system does not need to evolve toward its corresponding stable state. On the other hand, since the parameter steering process starts and ends at S_i , the system has sufficient time to reach a stable state there. In fact, in all simulations and experiments, we allow ample time at S_i to ensure this condition is met.

To clarify this critical point, we have revised the sentence in the main text from “...the system does not need to evolve toward its corresponding stable state” to “...the system—except at S_i —does not need to evolve toward its corresponding stable state”.

Based on these two points, the use of the term “CST” in the non-adiabatic case is questionable, as it appears to sacrifice the essential chiral and deterministic nature of

the adiabatic CST, and instead merely exhibits differences in the final state without a clear underlying mechanism or reliable predictability. Indeed, I suspect that the underlying mechanism of the observed “CST” in the non-adiabatic case is but the remnant of the adiabatic case, both related to the bistable spectrum.

Reply 5: We thank you for raising this important point. As discussed in **Reply 3**, to avoid confusion and to emphasize the different underlying mechanism, we have revised the term from 'CST' to 'chiral-like state transfer'. In that response, we also discussed the general trade-off between adiabatic evolution with full predictability and the practical need for faster evolution—an issue encountered in both EP-based CST and NEP-based NCST. From an experimental standpoint, it is often desirable to shorten the evolution time, even though this may compromise predictability to some extent, as long as CST or NCST can still be achieved. In the following, we will demonstrate that the proposed NCST scheme in our work remains robust over a broad parameter range.

Here we first clarify the underlying mechanism of the NCST under the non-adiabatic condition. The NCST discussed in our work originates from the contribution of the attractor properties for the stable states of nonlinear Hamiltonians. To ensure that the instantaneous state can fall into different basins of attraction as system evolves in different directions, the initial point S_i is required to lie within the bistable region. The regions where bistability can occur are governed by the landscape of the nonlinear system's eigen-spectrum. *Therefore, the underlying physical mechanism of NCST arises from the combined contribution of the attractor properties and the nonlinear eigen-spectrum landscape.* While, as you also noted, the role of the nonlinear eigen-spectrum landscape has been emphasized in the adiabatic regime^{7,9}, our work highlights the increasing significance of the attractor properties in the non-adiabatic cases. After all, even with an identical nonlinear eigen-spectrum, the chirality can be reversed by varying the steering speed.

Fig. R3 | Reliable predictability in non-adiabatic regimes. Dynamical evolution of the state under different parameter trajectories and evolution periods T . In (a, d, g), we fix $\delta y = 0.15$ and -0.15 for S_1 and S_2 , respectively. The values of δx for $\{S_1, S_2\}$ are set to be -0.04 in (a), -0.1 in (d) and -0.16 in (g). Panels (d, e, h) correspond to $T=50$ and (c, f, i) to $T=100$. Other parameters used are the same as those in Fig. 2.

Below, based on our experimental design, we show that our NCST scheme still maintains reliable predictability even in non-adiabatic regimes—that is, the NCST remains stable over a reasonably broad parameter range. Following the approach of typical experimental practice^{2,8}, we first specify representative parameter-evolution trajectories [see the first column of Fig. R3] that deviations may present during experiments due to technical limitations. We then vary the evolution time to analyze the final state and assess the feasibility of realizing non-adiabatic CST. The second and third columns of Fig. R3 show the evolution trajectory of steering parameters in the parameter space and the states in the phase space for different period (T). The different periods used here represent the maximum evolution-time deviations that may arise due to experimental imperfections. Clearly, NCST can be successfully realized within these non-adiabatic regimes, thereby demonstrating the reliable predictability required for

experimental implementation. In fact, for the path presented in Fig. R3(d), the NCST persists for $T > 50$, thereby demonstrating the robustness of our scheme in achieving NCST.

In summary, given the numerous similarities between NEP-based NCST and EP-based CST [see **Reply 3**], we use the term ‘nonlinear chiral-like state transfer (NCST)’ to highlight the distinction in the underlying mechanisms. From a practical perspective, reducing the evolution time is often desirable—provided that CST/NCST can still be successfully realized. While such reduction inevitably decreases the level of predictability to some extent, NCST in non-adiabatic regimes still exhibits *reliable predictability* over reasonably broad parameter ranges that are also well beyond typical experimental uncertainties.

To clarify this critical point, we have updated the terminology and included these discussions above as a new Note in the Supplementary Information (Note 5).

Another relevant comment is that, similar non-adiabatic process should also emerge in the Liouvillian case, though may not be easily observable in Rydberg gases due to the fast relaxation therein.

Reply 6: We totally agree with you that when the mean-field approximation holds well, a similar non-adiabatic process should also emerge in the Liouvillian case.

Finally, I believe that some of the additional results presented in the rebuttal letter—such as Fig. R1 and Fig. R3—are important for understanding the authors' claims. These figures should be included in the Supplementary Note to provide a more complete picture of the system's behavior.

Reply 7: We thank you for these constructive suggestions. Accordingly, we have included Fig. R1 and Fig. R3 from the previous response as two new notes (Note 3 and Note 4) in the Supplementary Information to provide a more comprehensive understanding of the system's behavior.

2 Response to the Second Reviewer -- NCOMMS-24-55305A

Reply 8: We acknowledge the co-review process and appreciate the thoughtful feedback provided through this collaborative effort. We thank all reviewers, including the Early Career Researcher, for your time and constructive comments that helped improve our manuscript.

3 Response to the Third Reviewer -- NCOMMS-24-55305A

The detailed derivation of the non-Hermitian nonlinear equations is what I looked for.
Recommend publication.

Reply 9: We sincerely thank you for carefully reviewing our manuscript and response and recommending this manuscript publication in Nature Communications.

References

1. Hassan, A. U. *et al.* Chiral state conversion without encircling an exceptional point. *Phys. Rev. A* **96**, 1–5 (2017).
2. Nasari, H. *et al.* Observation of chiral state transfer without encircling an exceptional point. *Nature* **605**, 256–261 (2022).
3. Li, A. *et al.* Hamiltonian Hopping for Efficient Chiral Mode Switching in Encircling Exceptional Points. *Phys. Rev. Lett.* **125**, 187403 (2020).
4. Shu, X. *et al.* Fast encirclement of an exceptional point for highly efficient and compact chiral mode converters. *Nat. Commun.* **13**, 2123 (2022).
5. Shu, X. *et al.* Chiral transmission by an open evolution trajectory in a non-Hermitian system. *Light Sci. Appl.* **13**, 65 (2024).
6. Uzdin, R., Mailybaev, A. & Moiseyev, N. On the observability and asymmetry of adiabatic state flips generated by exceptional points. *J. Phys. A Math. Theor.* **44**, 435302 (2011).
7. Wang, H., Assawaworrarit, S. & Fan, S. Dynamics for encircling an exceptional point in a nonlinear non-Hermitian system. *Opt. Lett.* **44**, 638 (2019).
8. Doppler, J. *et al.* Dynamically encircling an exceptional point for asymmetric mode switching. *Nature* **537**, 76–79 (2016).
9. Xie, C., Sun, K., Guo, G., Yi, W. & Xiang, G. Chiral switching of many-body steady states in a dissipative Rydberg gas. *arXiv:2402.02779v1*.

Response to the Reviewer Comments

We thank the reviewers for reviewing our manuscript, their comments, and constructive suggestions.

1 Response to the Frist Reviewer -- NCOMMS-24-55305B

The authors have updated the terminology to "nonlinear chiral-like state transfer" and included the corresponding explanations in the main text. As all my questions are clarified, I would like to recommend the work for publication in Nature Communications.

Reply 1: We sincerely thank you for carefully reviewing our manuscript and response and recommending this manuscript publication in Nature Communications.

Minor question:

Should the labels in Fig. S6 (l) and (o) be $S_1 S_2$ rather than $S_2 S_1$?

Reply 2: We sincerely thank you for pointing this out. We have corrected the label order in Fig. S6(l) and (o) to " $S_1 S_2$ " in the revised version.

2 Response to the Second Reviewer -- NCOMMS-24-55305B

Reply 1: We acknowledge the co-review process and appreciate the thoughtful feedback provided through this collaborative effort. We thank all reviewers, including the Early Career Researcher, for your time and constructive comments that helped improve our manuscript.

In the manuscript entitled "Chiral State Transfer Realized with a Minimal Set of Parameters," the authors study the physical mechanism of chiral state transfer (CST) near a third-order NEP (NEP3) in a nonlinear non-Hermitian system. They demonstrate CST without the prerequisite of adiabaticity through both theory and circuit experiments. They show that the CSTs in the system strictly arise from the basins of attraction for the stable states of nonlinear Hamiltonians. Furthermore, they simplify the trajectory of the steering process to three distinct points on a straight line in the parameter space and claim that they achieve CSTs with a minimal set of parameters. The work is well-organized and well-written, and it has the potential to generate significant interest in the field of non-Hermitian physics. I have the following concerns and comments, before I can make a recommendation.

1. The authors claim that they achieve CSTs with a minimal set of parameters, as the results shown in Fig. 2 (g-h) and Fig. 4. They simplify the trajectory of the steering process to three distinct points $\{S_i, S_1, S_2\}$ on a straight line in the parameter space. However, when referring to chiral state transfer (CST), it means that the final state after an encircling period depends only on the direction of encirclement, either clockwise or counterclockwise. Since the three points lie on a straight line, making the trajectory neither clockwise nor counterclockwise, it's difficult to claim the process chiral, rather than directional.

2. Since any random initial states will be quickly attracted to the 'nearest' stable modes, my question is whether the evolved state is always restricted to the stable state? If so, the final state should depend solely on the last step, i.e., S_1 to S_i or S_2 to S_i . In the first case, the final state always ends at the upper stable state, and in the second case, the final state always ends at the lower stable state.

3. Following the concern above, a relevant comment is that the observed 'CST' along the straight line is nothing but the hysteresis behavior across bistable transitions in nonlinear systems. Indeed, CST based on the Liouvillian exceptional structure (the open-system counterpart of the eigenmode landscape in Fig. 1b here) of Rydberg vapors has recently been reported (arXiv:2402.02779), wherein the CST along the straight line, as discussed here, is a direct result of the well-known bistability transition of the Rydberg gas.

4. The expression ' $g_A = 3/(1 + |\psi|) - 0.05$ ' in the caption of Fig. 1, should probably be ' $g_A = 3/(1 + |\psi|^2) - 0.05$ '.